# The Prevalence of Mild Cognitive Impairment in a Convenience Sample of 202 Gulf War Veterans

**DOI:** 10.3390/ijerph17197158

**Published:** 2020-09-30

**Authors:** Linda L. Chao

**Affiliations:** 1Department of Radiology and Biomedical Imaging, University of California, San Francisco, CA 94143, USA; linda.chao@ucsf.edu; 2Department of Psychiatry and Behavioral Science, University of California, San Francisco, CA 94143, USA; 3San Francisco Veterans Affairs Health Care System, San Francisco, CA 94121, USA

**Keywords:** mild cognitive impairment, Gulf War, posttraumatic stress, hippocampus, neuroimaging

## Abstract

Gulf War Illness (GWI) is a chronic, multisymptom disorder estimated to affect approximately 25–32% of Gulf War veterans (GWVs). Cognitive dysfunction is a common symptom of GWI. On the continuum of cognitive decline, mild cognitive impairment (MCI) is conceptualized as a transitional phase between normal aging and dementia. Individuals with MCI exhibit cognitive decline but have relatively spared activities of daily function and do not meet criteria for dementia. The current study sought to investigate the prevalence of MCI in a convenience sample of 202 GWVs (median age: 52 years; 18% female). Twelve percent of the sample (median age: 48 years) had MCI according to an actuarial neuropsychological criterion, a rate materially higher than expected for this age group. GWVs with MCI also had a smaller hippocampal volume and a thinner parietal cortex, higher rates of current posttraumatic stress disorder and major depressive disorder compared to GWVs without MCI. Because people with MCI are more likely to progress to dementia compared to those with normal cognition, these results may portend future higher rates of dementia among deployed GWVs.

## 1. Introduction

An estimated 250,000 Gulf War veterans (GWVs) are afflicted by a chronic health condition known as Gulf War Illness (GWI) or Chronic Multi-symptom Illness (CMI) [1,2]. GWI/CMI is associated with several concomitant symptoms including persistent fatigue, musculoskeletal pain, gastrointestinal and respiratory problems, skin rashes, and cognitive dysfunction [1]. Longitudinal studies suggest that most GWVs with GWI/CMI have not improved with time [3,4,5]; instead, many are getting worse [2]. A recent study suggests that GWVs have more chronic medical conditions and are developing these chronic medical conditions earlier than the general civilian population [6]. There have also been reports of a higher than expected incidence of cognitive impairment among GWVs [7,8,9,10]. One study found that at least half of the veterans from a population-based cohort of 1200 GWVs with and without GWI reported symptoms of cognitive dysfunction [8]. A recent meta-analysis concluded that Gulf War (GW) service is associated with impairment in three cognitive domains: attention and executive function, visuospatial ability, and learning and memory [9].

On the continuum of cognitive decline, impairment greater than that expected for one’s age but not severe enough to meet the criteria for dementia is classified as mild cognitive impairment (MCI) [11,12]. Individuals with MCI exhibit objective cognitive impairment, typically in one or two cognitive domains, but have relatively spared daily functioning abilities. Thus, they do not meet the criteria for dementia [13]. Research suggests that individuals with MCI are 3 to 5 times more likely to progress to any form of dementia compared to those with normal cognition [14,15,16,17], with an annual rate of progression of 12% in the general population and up to 20% in populations at higher risk [18].

Because memory is severely impaired in AD, historically MCI was characterized as an amnestic disorder [14]. The original criteria for MCI, proposed in 1999, include subjective memory complaint, objective memory impairment, preserved general cognitive function, intact activities of daily living, and not meeting criteria for dementia [14]. Subsequently, broader classification schemes of MCI were introduced that include non-amnestic forms of MCI and clinical subtypes that include single and multiple cognitive domains [19,20,21]. The conventional MCI criteria have remained essentially unchanged in the past 20 years, and some shortcomings with the original criteria have been noted. For example, subjective reports of memory decline can sometimes be more strongly related to emotional factors than to objective memory abilities [22]. In many large-scale research studies and clinical trials, the conventional MCI criteria have been operationalized to rely on impaired performance on a single measure of episodic memory (generally delayed memory for a one-paragraph story) to determine impaired memory functioning [23], which can be unreliable [24]. It has also been common for large-scale research studies and clinical trials to use performance on a screening measure of global cognition (e.g., the Mini Mental State Exam) as an indication of intact abilities in other cognitive domains, which can be insensitive for distinguishing MCI from cognitively normal individuals [25]. Reliance of the conventional MCI criteria on a clinician’s judgment of mild impairment based on a semi-structured clinical interview [23] can introduce variability across clinicians, sites, and timepoints. Finally, patients diagnosed with the conventional MCI criteria can sometimes revert back to normal cognitive status [26,27,28,29,30] and have heterogenous neuropsychological [31] and AD biomarker profiles [32].

In an attempt to make the MCI diagnosis more consistent and reliable, an actuarial method for diagnosing MCI was proposed in 2009 [33]. The actuarial method differs from the conventional method in the following ways: first, cognitive impairment is defined as performance that is one standard deviation (SD) below the age-adjusted norm on a neuropsychological test measure. In contrast, the conventional MCI criteria defines cognitive impairment as performance that is 1.5 SDs below the age-adjusted norm on a neuropsychological measure [14]. Second, the actuarial method requires *two* impaired scores within a cognitive domain because neurologically normal older adults can have high base rates of isolated low neuropsychological test scores [24]. In contrast, the conventional MCI criteria requires only one impaired score. Research suggests that patients classified as MCI by the actuarial neuropsychological criteria have stronger associations between cognitive function and hippocampal volume [34], higher levels of AD biomarkers (i.e., beta-amyloid, Aβ, total and hyperphosphorylated tau in cerebral spinal fluid, CSF), are more likely to possess the APOE ε4 allele, a genetic risk factor for AD, and more likely to convert to dementia than patients classified as MCI by the conventional criteria [35].

Most MCI studies have focused on individuals in their 70s [36]; however, it has been argued that MCI can be identified in adults before the age of 60 using neuropsychological assessments that do not have ceiling effects and cover multiple cognitive domains [37]. We previously reported that GWVs with subjective memory complaints have objective memory impairment [38]. Because this satisfies two of the original Petersen criteria for MCI [14], and because there is suggestive evidence that GWVs exhibit a higher than expected incidence of cognitive impairment [7,8,9,10] and may be experiencing accelerated aging [6], the current study sought to investigate whether GWVs exhibit a higher than expected prevalence of MCI. We also explored demographic, clinical, military, and deployment-related differences between GWVs with and without MCI.

## 2. Materials and Methods

### 2.1. Human Subjects

The study participants were 202 consecutive GWVs recruited from 2014 to 2018 at the San Francisco Veterans Affairs Health Care System (SFVAHCS). The parent study was approved by the Institutional Review Boards of the University of California, San Francisco (UCSF) and the SFVAHCS. All participants provided written informed consent prior to enrollment in the parent study. Because this study consisted of secondary analyses of pre-existing de-identified data, additional IRB approval was not required.

### 2.2. Measures

#### 2.2.1. Clinical Interview Assessments

Participants in the parent study were clinically screened with the following instruments: the Structured Clinical Interview for DSM-IV Diagnosis (SCID) [39], the Life Stressor Checklist-Revised [40], the Clinician Administered Posttraumatic Stress Disorder (PTSD) Scale (CAPS) [41], and the Ohio State University TBI Identification Method (OSU TBI-ID) Short Form [42]. See [43] for additional details about the clinical screening of this sample.

#### 2.2.2. Kansas Military History and Health Questionnaire

The Kansas Gulf War Military History and Health Questionnaire [44] was used to assess Kansas Gulf War Illness (GWI) case status [44], the Centers for Disease Control and Prevention (CDC) Chronic Multisymptom Illness (CMI) case status [45], and GW deployment-related experiences. See [43] for additional details about classification of GWI and CMC cases in this sample.

### 2.3. Classification of MCI

The present study sought to examine the prevalence of MCI in deployed GWVs. Although we previously reported evidence that GWVs with subjective memory complaints have objective memory impairment [38], which meets two of the original Petersen criteria for MCI [14], we could not use the conventional MCI criteria to classify GWVs because the parent study did not access activities of daily living or the clinical presence/absence of dementia. However, all participants in the parent study underwent neuropsychological assessment. Therefore, we used the actuarial neuropsychological criteria [33,46] to determine MCI status in the GWVs. The neuropsychological measures used to classify MCI are listed in Table 1.

In the original publication by Jak et al. describing the actuarial neuropsychological criteria [33], three neuropsychological measures in five cognitive domains were used to classify MCI. A subject was classified as MCI if s/he scored ≥1 SD below the age-adjusted norm on two measures in a cognitive domain (i.e., a subject with ≥2 scores ≥1 SD below the age-adjusted norm in the memory domain was classified as an amnestic MCI; a subject with ≥2 scores ≥1 SD below the age-adjusted norm in the executive function domain was classified as an executive function MCI). In a subsequent publication, Bondi et al. [35] used the actuarial neuropsychological criteria to classify subjects from the Alzheimer’s Disease Neuroimaging Initiative (ADNI). Because ADNI did not have as extensive a neuropsychological battery as that used in the original study by Jak et al. [33], the authors adapted the actuarial criteria to consider two neuropsychological measures in three cognitive domains for classifying MCI [35]. An ADNI subject was classified as MCI if s/he scored ≥1 SD below the age-adjusted norm on both measures in a cognitive domain (i.e., memory, executive function, or attention) or if s/he scored ≥1 SD below the age-adjusted norm on ≥1 measure in each of the three cognitive domains (i.e., a subject with one score 1 SD below the age-adjusted norm in all three domains was classified as MCI).

In this study, we used the same five cognitive domains describe by Jak et al. [33]. However, because we did not have three separate neuropsychological measures for the visuospatial and language domains, we employed a hybrid of the actuarial criteria described by Jak et al. [33] and Bondi et al. [35]: we considered two neuropsychological measures in three cognitive domains (i.e., episodic memory, executive function, and attention) and one neuropsychological measure in the other two cognitive domains (i.e., language and visuospatial function). A GWV was classified as MCI if s/he scored ≥1 SD below the age-adjusted norm on both measures in the episodic memory, executive function, or attention domain. A GWV was also classified as MCI if s/he scored ≥1 SD below the age-adjusted norm on ≥4 cognitive domains. A GWV was classified as having “intermediate” cognitive impairment if s/he scored ≥1 SD below the age-adjusted norm on ≤3 cognitive domains. GWVs with no score below the age-adjusted norm in any cognitive domain were classified as “cognitively normal” (CN). See Table 2.

### 2.4. Brain MRI

GWVs were scanned at the SFVSHCS on a 3 Tesla (T) Siemens Skyra MRI system equipped with a 32-channel receiver head coil. The MRI scan protocol included the following: T1-weighted 3D whole brain gradient echo MRI TR/TE/TI = 2500/2.98/1100 ms, 1.0 × 1.0 × 1.0 mm^3^ resolution and T2-weighted turbo spin echo MRI TR/TE 3200/11 ms, 0.9 × 0.9 × 3.0 mm^3^ resolution. The T2-weighted image was used to estimate intracranial volume (ICV). One hundred and ninety-four GWVs in the sample had artifact-free MRI data, from which we derived hippocampal volumes and measures of cortical thickness.

#### 2.4.1. Image Processing

Freesurfer Version 5.1 was used to label cortical and subcortical tissue classes and derive quantitative estimates of cortical thickness and regional brain volume [52,53,54]. To protect against type I error, and because we had no a priori hypotheses about laterality, volumes of the right and left hippocampus, and cortical thicknesses of the lobes of the brain and the insula were combined across the hemispheres to reduce the number of measurements. To further reduce the number of measures, we averaged across the Freesurfer parcels to examine cortical thickness of the frontal, temporal, parietal, and occipital lobes of the brain and the insula, as described in [55].

#### 2.4.2. Intracranial Volume (ICV) Measurement

After checking that all extracranial and skull structures were removed and all intracranial structures were fully preserved in the skull stripped images, the BET program (FMRIB Image Analysis Group, Oxford University, www.fmrib.ox.ac.uk/fsl) was used to determine intracranial volume (ICV) from the T2-weighted scan.

### 2.5. Statistical Analyses

Statistical analyses of the demographic, clinical, neuropsychological, and volumetric measures were conducted using IBM SPSS Statistics for Windows, version 26 (IBM Corp, Armonk, NY, USA). Demographic and descriptive characteristics were compared across three cognitive groups with analysis of variance (ANOVA) for continuous variables and chi-square tests for categorical variables. Because we previously reported significant effects of predicted exposure to the Khamisiyah plume on hippocampal and total cortical gray matter volumes [56,57,58], Khamisiyah exposure status was included as a covariate in analyses of the hippocampal volume and measures of cortical thickness. We also accounted for apolipoprotein E (APOE) genotype in analyses of hippocampal volume and cortical thickness. This is because the APOE ε4 allele is a genetic risk factor for sporadic AD [59,60,61], while the APOE ε2 allele is considered protective against AD [62,63]. The APOE ε4 allele has been associated with a smaller hippocampal volume [64,65] and thinner cortex [66] in cognitively normal individuals and in patients with AD [67] while the APOE ε2 allele has been associated with thicker cortex in cognitively intact individuals [68]. GWVs with APOE 3/4 or 4/4 genotype were classified as ε4 positive while GWVs with APOE 2/3 or 2/4 genotype were classified as ε2 positive. Because the APOE ε2 allele is considered protective against AD [62,63] GWVs with the 2/4 genotype were not classified as APOE ε4 positive.

#### 2.5.1. Covariates

Age [69,70], sex [71], and years of education [72,73] were included as covariates in analyses of hippocampal volume and cortical thickness because these variables have been shown to influence measures of brain volume and thickness. We also accounted for the presence or absence of Kansas GWI exclusionary condition (e.g., diabetes, heart disease other than hypertension, stroke, lupus, rheumatoid arthritis, cancer, liver and/or kidney disease) as a proxy measure of general overall health. ICV was included as a covariate in analysis of hippocampal volume to account for variations in head size between subjects that could reduce reliability [74]. Finally, we included any demographic and clinical variables that differed significantly between groups as covariates in the analyses.

#### 2.5.2. Correction for Multiple Comparisons

Because the Bonferroni correction for multiple comparisons can be overly conservative when applied to non-independent (i.e., correlated) measures, we adjusted the analyses of cortical thickness of the four lobes of the brain and insula according to the number of regions analyzed (*n* = 5) and the average intercorrelations among the regions [75]. With an average intercorrelation *r* = 0.58, a 2-sided adjusted *p* = 0.03 was considered statistically significant. Similarly, analyses of GW deployment-related exposures were adjusted for multiple comparisons according to the number of exposures analyzed (*n* = 15) and the average intercorrelations among the exposures [75]. With an average intercorrelation *r* = 0.21, a 2-sided adjusted *p* = 0.006 was considered statistically significant.

#### 2.5.3. Post-Hoc Analyses

We used hierarchical linear regression to examine the ability of current PTSD, current MDD, and history of alcohol abuse/dependence (independent variables entered in the last step of the regression model) to predict MCI status (dependent variable) over and above demographic and military characteristics that differed significantly between the groups (independent variables entered into the first step of the regression model). We used also hierarchical linear regression to examine the ability of MCI status (independent variable entered in the last step of the regression model) to predict hippocampal volume (dependent variable in second post-hoc analysis) and parietal cortex thickness (dependent variable in third post-hoc analysis) over and above demographic and clinical variables (i.e., age, sex, years of education, current PTSD and MDD, history of alcohol abuse/dependence, ICV, APOE ε4 and ε2 status, CDC CMI, and Khamisiyah plume exposure status, entered into the first step of the regression models). In a final post-hoc stepwise linear regression analysis, we examined the ability of all the demographic, clinical, and deployment-related exposures that differed significantly between the groups to predict MCI status (dependent variable).

## 3. Results

Twelve percent of the GWVs (*n* = 25) in the sample were classified as MCI according to the actuarial neuropsychological method, while 39% (*n* = 79) were classified as “cognitively normal” (CN, i.e., had no impaired scores in any cognitive domains). The rest of the GWVs (49%, *n* = 98) were classified as “intermediate.” Five of the intermediate GWVs had impaired scores (i.e., >1 SD below the age-corrected norm) in 3 cognitive domains; 25 had impaired scores in 2 cognitive domains; 68 had impaired scores in 1 cognitive domain.

Table 3 summarizes the demographic and clinical characteristics of the three cognitive groups. The MCI group tended to be younger (F_2.201_ = 2.49, *p* = 0.09) and have fewer years of formal education (F_2.201_ = 2.49, *p* = 0.09) compared to the other two groups. Significantly more GWVs with MCI were enlisted personnel during the GW (96%, χ^2^ = 13.32, *df* = 2, *p* = 0.003), had current PTSD (40%, χ^2^ = 17.12, *df* = 2, *p* = 0.001) and current MDD (24%, χ^2^ = 7.30, *df* = 2, *p* = 0.03), and a history of alcohol abuse/dependence (44%, χ^2^ = 6.73, *df* = 2, *p* = 0.04) compared to the other groups. There were significantly more CDC CMI cases, both mild–moderate (80%, χ^2^ = 6.52, *df* = 2, *p* = 0.04) and severe (40%, χ^2^ = 12.89, *df* = 2, *p* = 0.002), in the MCI than the other groups. There were no differences in Kansas GWI cases or in the number of GWVs with Kansas GWI exclusionary conditions. The CN group had fewer African Americans compared to the other two groups (1% versus 11% in the intermediate and 24% in the MCI, χ^2^ = 16.10, *df* = 2, *p* = 0.001).

In post-hoc analysis, we used hierarchical linear regression to examine the ability of the clinical variables (i.e., current PTSD, current MDD, and history of alcohol abuse/dependence, independent variables entered in the last step of the regression model) to predict MCI status (dependent variable) over and above demographic and military characteristics that differed significantly between the groups (i.e., race and rank during the GW, entered as independent variables in Step 1 of the model). The post-hoc regression models were significant (*p’s* ≤ 0.03, 0.08 ≤ R^2^ ≤ 0.11). In the first step of the model, rank (i.e., enlisted, standardized coefficient *β* = 0.21, *t* = 2.99, *p* = 0.003) and race (i.e., non-white, standardized coefficient *β* = 0.21, *t* = 3.04, *p* = 0.003) were significantly associated with MCI status. Even after accounting for rank and race in the first step of the model, current PTSD status (standardized coefficient *β* = 0.17, *t* = 2.47, *p* = 0.01) was significantly associated with MCI status in the second step of the model. Current MDD and history of alcohol abuse/dependence were not significantly associated with MCI status.

Table 4 summarizes the neuroimaging results. The MCI group had a smaller hippocampal volume compared to the other groups, even after accounting for ICV, age, sex, education, rank during the GW, current PTSD and MDD diagnoses, history of alcohol abuse/dependence, CDC CMI case status, predicted exposure to the Khamasiyah plume, Kansas GWI exclusionary conditions (as a proxy measure for general overall health) and APOE ε2/ε4 genotype (F_2.189_ = 6.76, *p* = 0.001). Planned contrasts revealed that the MCI group had a smaller hippocampal volume compared to CN (*p* < 0.001) and intermediate (*p* = 0.001) groups, but there were no significant hippocampal volume differences between intermediate and CN groups. The MCI group also had a thinner parietal cortex compared to the intermediate and CN groups, even after accounting for age, sex, education, rank during the GW, current PTSD and MDD diagnoses, history of alcohol abuse/dependence, CDC CMI case status, predicted exposure to the Khamasiyah plume, Kansas GWI exclusionary status, and APOE ε2/ε4 genotype (F_2,181_ = 4.10, *p* = 0.018). Planned contrasts revealed that the MCI group had a thinner parietal cortex compared to CN (*p* = 0.005) and intermediate (*p* = 0.03) groups, but there were no significant parietal cortex thickness differences between intermediate and CN groups. Although there was a group difference in temporal cortex thickness (F_2,183_ = 3.29, *p* = 0.04), this difference was not significant after adjustments for multiple comparisons according to the number of regions analyzed (*n* = 5) and the average intercorrelations among those regions [75]. With an average intercorrelation of *r* = 0.58 between the thickness of the four lobes of the brain and insula, a 2-sided adjusted *p* = 0.03 was considered statistically significant.

In post-hoc analyses, we used hierarchical linear regression to examine the ability of MCI status (independent variable entered in the last step of the model) to predict hippocampal volume and parietal cortex thickness (dependent variables) over and above demographic and clinical variables. All fits in the post-hoc regression for hippocampal volume were significant (*p’s* ≤ 0.002, 0.30 ≤ R^2^ ≤ 0.33). After accounting for ICV (standardized coefficient *β* = 0.51, *t*= 7.40, *p* < 0.001), age (standardized coefficient *β* = −0.20, *t*= −3.01, *p* = 0.001), and Khamisiyah exposure status (standardized coefficient *β* = −0.13, *t*= −1.99, *p* < 0.05) in the first step of the model, MCI status was still significantly associated with hippocampal volume (standardized coefficient *β* = 0.20, *t*= 3.07, *p* = 0.002). In the post-hoc regression for parietal cortex thickness, age was inversely related to parietal cortex thickness (standardize coefficient *β* = −0.20, *t*= −2.44, *p* = 0.016); however, the first model with demographic and clinical variables was not statistically significant (R^2^ = 0.04, p = 0.09). Only the second model was significant (R^2^ = 0.07, p = 0.008) because MCI status was significantly associated with parietal cortex thickness (standardized coefficient *β* = 0.21, *t*= 2.67, *p* = 0.008).

Table 5 summarizes the group differences in GW deployment-related exposures and experiences. More GWVs in the MCI group reported witnessing smoke from burning oil well fires, hearing chemical alarms sound, coming into contact with dead animals, using powdered pesticides directly on skin, witnessing their living areas being sprayed with pesticides, and coming into contact with chemical agent resistant coating (CARC) paint compared to the other groups (see Table 5). However, after adjustment for multiple comparisons according to the number of exposures (*n* = 15) and the average intercorrelation among the exposures (r = 0.21) [75], only differences in the frequency of contact with dead animals and witnessing living areas being sprayed with pesticides remained significant.

In a final post-hoc analysis, we examined the relationship between MCI status (dependent variable) and demographic and military characteristics (race and rank during GW), clinical characteristics (current PTSD and MDD, history of alcohol abuse/dependence, and CDC CMI status) and deployment-related experiences (coming into contact with dead animals and witnessing living area sprayed with pesticides) that differed significantly between the groups. The stepwise linear regression models (*p*’s ≤ 0.01, 0.07 ≤ R^2^ ≤ 0.15) revealed three significant predictors of MCI status: race (being non-white, standardized coefficient *β* = 0.20, *t* = 2.82, *p* = 0.005), current PTSD diagnosis (standardized coefficient *β* = 0.20, *t*= 2.94, *p* = 0.004), and rank (enlisted during the GW, standardized coefficient *β* = 0.17, *t*= 2.48, *p* = 0.01). GW deployment-related experiences were not significantly associated with MCI status after accounting for these three variables.

## 4. Discussion

The main finding of this study is that 12% of GWVs in a convenience sample originally recruited for a VA-funded study on the effects of predicted exposure to the Khamisiyah plume on brain structure and function had MCI according to an actuarial neuropsychological criteria [33,35]. MCI, conceptualized as the transition between normal cognitive aging and dementia [76], is widely considered to be a prodromal condition indicative of future dementia such as AD [76]. Although not every patient with MCI progresses to dementia [14,77], it is noteworthy that GWVs with MCI had smaller hippocampal volume and thinner parietal cortex, two hallmarks of AD pathogenesis [78,79] compared to GWVs without MCI. The group difference in hippocampal volume and parietal cortex thickness remained significant even after accounting for demographic and clinical variables, predicted exposure to the Khamisiyah plume, APOE genotype, and general overall health, approximated by using Kansas GWI exclusionary status.

How significant is a prevalence of 12% of MCI in GWVs who had a median age of 48 years at the time they were studied? According to the American Academy of Neurology Practice update summary of MCI [80], the prevalence of MCI in the general population is 6.7% for ages 60–64, 8.4% for 65–69, 10.1% for 70–74, 14.5% for 75–79, and 25.2% for ages 80–84. Very few research studies of MCI focus on adults under the age of 60 [36]. The few studies that have reported prevalence in the range of 0%−13% [81,82,83]. If the higher estimates of MCI in adults under 60 is valid, then our finding that 12% of deployed GWVs had MCI is not out of the ordinary. In a study of 1126 twins from the Vietnam Era Twin Study of Aging (VETSA) who were 51–59 years old at the time of study, Kremen et al. reported a prevalence of 1%−65% for MCI [37]. However, it seems unlikely that 65% of Vietnam veterans in their 50s would have MCI given that the prevalence of MCI increases with age [84] and 65% is significantly higher than the prevalence of MCI in octogenarians, estimated to be 25% by the American Academy of Neurology [80]. Consequently, Kremen et al. proposed that lower rates of MCI are more likely to be valid in adults under 60 [37]. If we assume that Kremen et al.’s proposal is correct, and that the prevalence of MCI in adults under 60 is lower than the prevalence of MCI in people aged 60–64, estimated to be 6.7% by the American Academy of Neurology [80], then 12% MCI in deployed GWVs is almost twice the prevalence of MCI expected in the general population.

Two case definitions for GWI have been endorsed by the Institute of Medicine for use in clinical diagnosis and research investigations [1]: the CDC CMI definition [45] and the Kansas GWI definition [44]. Although cognitive dysfunction is a symptom of both, it is interesting that was no difference in the rate of Kansas GWI between the three cognitive groups. In contrast, there were significantly more CDC CMI cases, both severe and mild–moderate cases, in the MCI group. This is likely because the Kansas GWI definition requires cases to have multiple chronic symptoms in at least three of six categories and cognitive difficulties falls within the broader category of neurological/cognitive/mood symptoms. In contrast, the CDC CMI definition requires only symptoms in two of three categories and the mood-cognition category captures many symptoms common in individuals with MCI (i.e., difficulty remembering or concentrating, word finding difficulties, feeling depressed, feeling moody, feeling anxious, trouble sleeping).

The mechanisms underlying higher rates of MCI in GWVs are likely to be complex and interlinking with no single process explaining the relationship. However, post-hoc regression analysis revealed a strong association between current PTSD and MCI status, even after accounting for demographic characteristics. PTSD is a stress-related condition that develops in some individuals after experiencing a traumatic event [85]. It has been postulated that dysregulation of the hypothalamic-pituitary-adrenal (HPA) axis occurs in some people following exposure to severe trauma [86]. Chronic hyperactivation of the HPA axis may lead to aberrant neuroimmune responses [87], which, in turn, can result in damage to the hippocampus [88,89,90,91]. A large (*n* = 1868) consortium study recently confirmed the negative association between PTSD and hippocampal volume [91]. Notably, hippocampal atrophy is also a pathological hallmark of AD [92]. There is also suggestive evidence that PTSD may induce epigenetic changes that disrupt a number of physiological mechanisms/systems such as the metabolic, immune, and inflammatory systems, which renders individuals with PTSD vulnerable for developing a variety of co-morbid chronic diseases [93]. Aberrant immune responses may also interrupt anti-inflammatory Aβ clearance mechanisms that creates a switch towards pro-inflammatory mechanisms which could cause neuronal necrosis [94]. One consequence of this would be cognitive impairment. Indeed, PTSD has been associated with impaired cognition [95,96,97,98,99], increased risk for dementia [94,99,100], and increased Aβ burden [101], but see [102].

Although there were higher rates of current MDD among GWVs with MCI compared to the other groups (24% vs. 8% and 6% in intermediate and CN), post-hoc analysis suggested that current MDD was not significantly associated with MCI status in the present study. Nevertheless, it is noteworthy that MDD has been associated with impaired cognitive function [103,104,105,106]. Furthermore, there is a large body of literature suggesting that depression in late life [107,108,109,110,111] as well as middle age [112,113] increases the risk of developing cognitive impairment and dementia. An epidemiological study of more than 13,000 subjects found that the risk of developing AD was approximately doubled in individuals with late-life depressive symptoms while the risk of vascular dementia was more than tripled in those with both mid- and late-life depression [114]. Neuropathology studies have linked history of depression with increased amyloid plaques and neurofibrillary tangles, two neuropathological hallmarks of AD [115,116]. Prolonged damage to the hippocampus due to hypercorisolaemia may play a role linking both PTSD and MDD to cognitive impairment and risk for dementia [100,117].

Consistent with epidemiological findings that substance use disorders are commonly comorbid with PTSD [118,119,120,121] and MDD [122,123], there was a higher prevalence of history of alcohol abuse/dependence in MCI group compared to the other two groups (44% vs. 19% and 23% in intermediate and CN). Although post-hoc analysis suggested that history of alcohol use disorder was not significantly associated with MCI status in the present study, like PTSD and MDD, heavy alcohol use has been associated with deleterious effects on the hippocampus [124] and cognitive function [125,126,127,128]. Future studies will be necessary to ascertain the relationship between PTSD, MDD, history of alcohol abuse/dependence and MCI in GWVs.

Contact with dead animals during the GW has been considered a proxy for contact with chemical warfare agents [129]. There have been reports of negative associations between self-reported exposure to pesticides and cognitive performance in non-demented individual who live in areas near pesticide-sprayed fields [130]. In the current study, GWVs with MCI reported higher frequencies of coming into contact with dead animals and seeing their living area being sprayed or fogged with pesticide during the GW; however, neither deployment-related exposure was significantly associated with MCI status after accounting for race, rank, and PTSD in post-hoc regression analyses.

The current study also found a significant association between race (i.e., being non-white) and MCI status. This is consistent with the reports of higher rates of MCI and dementia in African Americans [131,132,133,134,135]. In fact, it has been suggested that African Americans and Hispanics may be more likely to develop AD and other dementias than their non-Hispanic White counterparts [136,137] because of differences in underlying risk factors [136,137,138,139]. There were also significantly more enlisted personnel in the MCI group compared to the other two cognitive groups. In the U.S. Armed Forces, military rank is commonly considered a proxy for socioeconomic status (SES) because higher SES correlates with higher rank and there is a direct relationship between military rank and annual income [140]. Many studies have found higher risk of AD and other dementias [141,142,143,144,145,146] and higher rates of MCI [82,146,147] among individuals with lower SES. There is a well-established relationship between higher SES levels and cognitive function, particularly for executive function and language tasks [148]. One explanation may be that higher cognitive function leads to more intellectual and higher paying occupations, and greater wealth affords enriching lifestyles than can contribute to cognitive reserve [149], which may confer protection against the effects of neurodegeneration [150].

The findings of this study should be considered in the context of some limitations: first, this study had a cross-sectional, non-random, non-experimental design because the parent study from which data for the secondary analyses were derived was not originally designed to investigate the relationship between GW deployment and the prevalence of MCI. Thus, the current findings may not accurately reflect the true prevalence or severity of MCI among GWVs and cannot determine causal links. For example, history of TBI with loss of consciousness (LOC) has been associated with an approximately 2.5 year earlier diagnosis of MCI [151]. However, TBI with LOC was exclusionary in the parent study. Further, the parent study excluded participants with a lifetime history of psychotic or bipolar disorders and/or drug abuse or dependence in the past 12 months, which may have biased the current findings. Second, MCI was defined using an actuarial neuropsychological criteria [33] and the GWVs did not undergo neurological evaluations. Nevertheless, it is worth noting that MCI patients identified with the actuarial neuropsychological criteria have been shown to exhibit significant cerebral spinal fluid AD biomarker associations, more stable diagnoses, and have a larger percentage of individuals who progressed to dementia than MCI patients diagnosed by the conventional MCI criteria [35]. Third, the present study used a hybrid of the actuarial neuropsychological criteria described by Jak et al. [33] and Bondi et al. [35]. The original description of the actuarial neuropsychological criteria employed three neuropsychological measures in five cognitive domains. Patients were categorized as MCI if they had impaired scores on two measures in a cognitive domain [33]. In a subsequent publication, the actuarial neuropsychological criteria was adapted for the ADNI sample and two neuropsychological measures in three cognitive domains were used [35]. ADNI subjects were classified as MCI if they had an impaired score on two measures in one cognitive domain or one impaired score in three cognitive domains. Because the neuropsychological battery used in the parent study only had one measure in the visuospatial domain and one measure in the language domain, we modified the actuarial neuropsychological criteria to categorize GWVs as MCI if they had two impaired scores in the cognitive domains with two measures (i.e., memory, executive function and attention), or one impaired score in at least four of the five domains. Therefore, it may be possible that some of the GWVs classified as “intermediate” in the present study may, in fact, have MCI. Finally, we did not have information about the veterans’ premorbid cognitive abilities (e.g., from the Armed Forces Qualification Test). Previous research has shown that low premorbid cognitive abilities may be a risk factor for MCI, whereas high premorbid cognitive abilities may be protective against MCI [151].

## 5. Conclusions

Assuming that the prevalence of MCI in people under 60 is lower than the prevalence of MCI in people aged 60–64, estimated to be 6.7% [80], the finding that 12% of GWVs (median 48 years at the time of testing) had MCI is nearly twice the prevalence rate of MCI expected in the general population. This finding is consistent with the idea that GWVs are aging at a faster rate than the general population [6]. Furthermore, GWVs with MCI had hippocampal atrophy and a thinner parietal cortex, two hallmarks of AD pathogenesis [78,79], compared to GWVs without MCI. Because individuals with MCI develop dementia at a higher rate than the general population (10–15% versus 1–2% per year) [12], if these results are confirmed in a larger, more general sample of GWVs, it may portend higher rates of future dementia in deployed GWVs. With the advent of in vivo biomarkers of amyloid and tau, AD is increasingly being conceptualized as a biomarker-driven diagnosis rather than a clinical syndrome [152]. Therefore, it will be informative to examine in vivo levels of amyloid and tau in GWVs with MCI in future studies.

## Figures and Tables

**Table 1 ijerph-17-07158-t001:** Neuropsychological measures used in the actuarial neuropsychological criteria for classifying MCI.

Domain	Measures
Episodic Memory	California Verbal Learning Test-II [47] (CVLT) Trials 1–5 total recall CVLT long-delay free recall
Executive Function	Trail Making Test [48] (TMT) Part B Delis–Kaplan Executive Function System [49] (D-KEFS) Color-Word Inference Test, color-word inhibition condition
Attention	Weschler Adult Intelligence Scale-III [50] (WAIS-III) Digit Span TMT [48] Part A
Language	Boston Naming Test [51]
Visuospatial Function	WAIS-III [50] Block Design

**Table 2 ijerph-17-07158-t002:** Operationalization of MCI, intermediate, and cognitively normal status.

MCI	Score ≤ 1 SD Below the Norm on both Measures in Episodic Memory Domain or Executive Function Domain or Attention Domain or Score ≤ 1 SD Below the Norm on ≥1 Measure in ≥4 Cognitive Domains
Intermediate cognitive impairment	Score ≤ 1 SD below the norm on ≤3 cognitive domains
Cognitively normal	No score < 1 SD below the norm in any cognitive domains

**Table 3 ijerph-17-07158-t003:** Demographic and clinical characteristics in the entire study sample and by cognitive status.

	All	MCI	Intermediate	CN	Statistics
*N*	202	25 (12%)	98 (49%)	79 (39%)	
Age (years)	54.1 (7.7)	51.8 (6.7)	53.7 (7.3)	55.4 (8.4)	F = 2.49
Education (years)	15.6 (2.3)	14.8 (2.2)	15.5 (2.3)	16.0 (2.3)	F = 2.49
Male	166 (82%)	22 (88%)	76 (78%)	68 (86%)	χ^2^ = 2.83
Race					χ^2^ = 16.10 **
Caucasian	148 (73%)	15 (60%)	66 (67%)	67 (85%)	
African American	18 (9%)	6 (24%)	11 (11%)	1 (1%)	
Other	36 (18%)	4 (16%)	21 (21%)	11 (14%)	
Enlisted personnel during GW	150 (74%)	24 (96%)	77 (79%)	49 (62%)	χ^2^ = 13.32 **
Military service during GW					χ^2^ = 1.71
Active duty	155 (77%)	18 (72%)	78 (80%)	59 (75%)	
Reserves	38 (19%)	6 (24%)	15 (15%)	17 (22%)	
National guard	9 (5%)	1 (4%)	5 (5%)	3 (4%)	
Branch of Service during GW					χ^2^ = 0.72
Army	126 (62%)	17 (68%)	61 (62%)	48 (61%)	
Marines	39 (19%)	4 (16%)	20 (20%)	15 (19%)	
Navy	20 (10%)	2 (8%)	9 (9%)	9 (11%)	
Air force	17 (8%)	2 (8%)	8 (8%)	7 (9%)	
Current PTSD	28 (14%)	10 (40%)	12 (12%)	6 (8%)	χ^2^ = 17.12 ***
Current MDD	19 (9%)	6 (24%)	8 (8%)	5 (6%)	χ^2^ = 7.30 *
Hx ETOH abuse/depend.	48 (24%)	11 (44%)	19 (19%)	18 (23%)	χ^2^ = 6.73 *
Hx substance abuse/depend.	16 (8%)	4 (16%)	5 (5%)	7 (9%)	χ^2^ = 3.40
Psychotropic medication use	40 (20%)	9 (36%)	18 (18%)	13 (16%)	χ^2^ = 4.81
TBI Hx					χ^2^ = 9.02
Improbable	105 (52%)	13 (52%)	50 (51%)	42 (53%)	
Possible	49 (24%)	4 (16%)	20 (31%)	15 (19%)	
Mild	46 (23%)	8 (32%)	16 (16%)	22 (28%)	
Moderate	2 (1%)	0 (0%)	2 (2%)	0 (0%)	
APOE genotype ^a^					χ^2^ = 6.11
ε2/ε3	21 (11%)	3 (12%)	12 (13%)	6 (8%)	
ε2/ε4	8 (4%)	2 (8%)	3 (3%)	3 (4%)	
ε3/ε3	129 (65%)	15 (60%)	58 (60%)	56 (73%)	
ε3/ε4	33 (16%)	5 (20%)	18 (19%)	10 (13%)	
ε4/ε4	7 (4%)	0 (0%)	5 (5%)	2 (3%)	
Mild–moderate CDC CMI cases	126 (62%)	20 (80%)	64 (65%)	43 (63%)	χ^2^ = 6.52 *
Severe CDC CMI cases	36 (18%)	10 (40%)	19 (19%)	7 (9%)	χ^2^ = 12.89 **
Kansas GWI cases	83 (41%)	14 (56%)	41 (42%)	28 (35%)	χ^2^ = 3.36
Kansas exclusionary condition	59 (29%)	7 (28%)	31 (32%)	21 (27%)	χ^2^ = 0.56
Predicted Khamisiyah exposure	88 (44%)	11 (44%)	44 (45%)	33 (42%)	χ^2^ = 0.18

* *p* < 0.05, ** *p* < 0.01, and *** *p* < 0.001. ^a^ data unavailable for two intermediate and two CN (cognitively normal) veterans.

**Table 4 ijerph-17-07158-t004:** Neuroimaging results.

	MCI	Intermediate	CN	Statistics
Hippocampal volume (cc)	8.05 (0.98)	8.59 (0.80)	8.68 (0.79)	F = 6.76 ^a^
Cortical Thickness (mm)				
Frontal lobe	2.54 (0.09)	2.59 (0.10)	2.59 (0.10)	F = 2.28
Parietal lobe	2.33 (0.11)	2.38 (0.09)	2.40 (0.09)	F = 4.10 ^b^
Temporal lobe	2.87 (0.11)	2.95 (0.12)	2.95 (0.11)	F = 3.42 ^†^
Occipital lobe	1.90 (0.11)	1.92 (0.10)	1.95 (0.10)	F = 2.07
Insula	2.98 (0.15)	3.03 (0.13)	3.06 (0.14)	F = 2.13

^a^ ANCOVA with ICV, age, sex, education, rank, CMI case status, Kansas GWI exclusionary status, predicted Khamisiyah exposure status, current PTSD, current MDD, history of ETOH abuse/dependence, and APOE ε2/ε4 status as covariates. ^b^ ANCOVA with age, sex, education, rank, CMI case status, Kansas GWI exclusionary status, predicted Khamisiyah exposure status, current PTSD, current MDD, history of ETOH abuse/dependence, and APOE ε2/ε4 status as covariates. ^†^
*p* = 0.04, not significant after correction for multiple comparisons.

**Table 5 ijerph-17-07158-t005:** Deployment-related exposure or experience as a function of cognitive group.

	Did Not Experience			Experienced for 1–6 Days			Experienced for 7–30 Days			Experienced for >30 Days			χ2	*p*
	MCI	Int	CN	MCI	Int	CN	MCI	Int	CN	MCI	Int	CN		
Smoke from oil well fires	4	14	29	16	20	14	20	34	29	60	32	28	16.51	0.01
Heard chemical alarms	12	25	27	28	37	39	24	27	28	36	11	6	16.13	0.01
Within 1 mild of SCUD missile explosion	29	55	56	58	33	33	8	8	9	4	4	3	6.98	0.32
Contact with POWs	48	45	52	28	28	30	12	19	11	12	8	6	3.39	0.76
**Contact with dead animals**	**32**	**63**	**62**	**44**	**16**	**32**	**8**	**14**	**5**	**16**	**6**	**1**	**22.62**	**0.001**
Contact with destroyed enemy vehicles	20	35	39	44	30	34	20	27	22	16	9	5	6.78	0.34
Contact with vehicles destroyed by friendly fire	60	76	82	28	18	16	4	3	1	8	3	1	6.43	0.38
Cream/liquid pesticides	28	36	51	8	5	9	16	21	13	48	38	27	8.72	0.19
Powdered pesticide	60	86	89	8	3	1	4	4	4	28	6	6	16.22	0.01
Pesticide-treated uniform	58	67	64	0	2	8	13	9	8	29	22	21	5.69	0.46
Wore flea collar	88	95	96	0	1	0	0	2	3	12	2	1	9.65	0.14
**Living area sprayed with pesticides**	**52**	**78**	**76**	**8**	**9**	**13**	**12**	**8**	**8**	**28**	**5**	**4**	**18.86**	**0.004**
Took PB pills	20	26	23	36	22	26	24	25	28	20	28	23	2.9	0.82
Contact with CARC paint	44	77	72	16	11	10	12	4	9	28	8	9	13.37	0.04
Lived in tent with fuel burning heater	24	39	46	8	5	9	12	7	10	56	49	35	7.04	0.32

Int: Intermediate cognitive impairment; POWs: prisoners of war; PB: pyridostigmine. Values are %; Bold type indicates significant after adjustment for multiple comparisons.

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
