# Peer review of "The Prevalence of Mild Cognitive Impairment in a Convenience Sample of 202 Gulf War Veterans"

_ijerph, 2020, doi:10.3390/ijerph17197158_

Round 1

Reviewer 1 Report

The present paper assesses the incidence of Mild Cognitive Impairment (MCI) among Gulf War Veterans (GWV) and tries to compare these findings with similar studies on community samples of a comparable age range. Since MCI is regarded as a prodromal condition to dementia, these findings would help in elaborating the potential role of environmental factors and external stressors (as those experienced in the Gulf War) in MCI and dementia later on.

Herein lies the main strength of the paper, since it assesses a multitude of factors, like the exposure to potentially harmful chemical compounds, exposure to traumatic events, the presence of mental disorders, but also genetic and neuroimaging factors in relation to MCI.

However, these factors are approached and analysed in a way that is rather inconclusive, even misleading. Further, the structure of the paper with a very short introductory section and a large discussion section, where a multitude of issues are addressed and discussed that weren’t initially mentioned in the aims of the paper, leads to some confusion about what this paper was really about. In my opinion, this article has many strengths and a great potential, which the author seems reluctant to approach.

The major issue of concern is the main goal of the paper (the prevalence of MCI in GWV) and the way it is displayed and discussed. The author compares the obtained results (prevalence rates of 12%) with those of other studies which were conducted on similarly aged individuals from the general population. These rates appear similar, ranging from 0%-13%, or even higher, ranging from 1%-65%. Instead of discussing the possibility of GWV having similar rates of MCI as the rest of the population, the author argues in the abstract section of the manuscript that a MCI prevalence of 12% “is significantly higher than the estimated rate of MCI among individuals 45-65 years old in the general population” (l. 19-20). In the abstract section within the submission form, the text is different: “This is significantly higher than the estimated 0.1% of MCI among individuals 45-65 years old in the general population.”. It is not evident how the author arrives at the aforementioned rate of 0.1%. The author seems to supports this argument by stating that the results from the other studies are regarded as not reliable and the prevalence rates overestimated (l. 220-227). The author arrives finally at the following conclusion: “The present study found a higher rate of MCI among GWVs who were twenty years younger than average age of patients with MCI [58]” (l. 340–341). No data are provided with regard to the prevalence rates in these older individuals to justify this rather perplexing statement and some readers may assume that these rates are actually lower. However, in the referred paper, these rates range between 4-42%, while the present paper reports a prevalence of 12%. The author then suggests that: “This findings supports the idea that GWVs are aging at a faster rate than the general population.” (l. 341-342). To arrive to such a conclusion seems not justified, even more because the author found that “The MCI group tended to be younger (F2,201 = 2.49, p=0.09)” (l.164).

Secondly, the author tries to access causality of MCI among GWV with regard to the effect of mental disorders and deployment-related exposure to chemicals and air pollutants. The author concludes that all these conditions are probable risk factors for MCI in GWV. This argumentation would have been more solid if the author had performed regression analysis, taking all significant covariates into account, such as demographic data. The author clearly performed regression analysis with regard to MCI and neuroimaging data, so it is a surprise that the author didn’t chose to proceed the same way here.

Some minor issues:

Introduction:

This study has more goals than to simply assess the prevalence of MCI among GWV and this has to be addressed clearly. Further, there are only few studies on the prevalence of MCI in individuals aged <65 years, which I think deserves more emphasis in the introduction, since that would underline the potential usefulness of this paper. I also recommend the use of the term “prevalence” instead of “incidence”, since the term “incidence” refers to the rate of occurrence of new cases at a given time.

Materials and Methods:

l. 78-83: Classification criteria need to be described in more detail. The author states that “Jak et al. [27] used three neuropsychological measures…”, while “Bondi et al. [28] used two neuropsychological measures…”. More information of how exactly these measures are “used” would be helpful.

l. 146-155: 15 different exposures are displayed, however only 4 appear in statistical analysis (tables 4&5). This applies of course to the correction for multiple comparisons (l. 154-155). If only 4 different exposures were taken into account, then n=4 and not 15. And what about these other 11 types of exposure? Some of them refer to acute stress reactions and exposure to life threatening events, like for example, “being within 1 mile of a SCUD missile explosion”. Why didn’t the author make any comments on that? It would be very interesting to see in which degree some traumatic events predispose an individual to MCI and if this connection vanishes after controlling for PTSD. Do neurophsysiological mechanisms underlying an acute stress reaction have any effect of MCI prevalence and on hippocampal atrophy later in life? Are these effects rather mediated by the neurophysiological mechanisms involved in PTSD? From a clinical point of view, I would prefer to read more about that and less about actuarial vs conventional diagnostic criteria, which seems to cover a lot of space in the present manuscript.

Results:

Table 2 needs some attention: There can’t be 7 (8%) individuals of Kansas GWI cases in the MCI group. 7 (28%) seems accurate.

With regard to ANCOVA (Table 3), 95% confidence intervals have to be provided, in order to draw additional conclusions about the reliability of the displayed results.

l. 205-207. As mentioned above, this statement needs to be revised, since there are only 4 factors involved which have to be taken into account when adjusting to multiple comparisons.

Discussion:

l. 218-227. Paragraphs like these appear usually the Introduction, since they summarize previous work that has been accomplished, leading to argumentations about the necessity of conducting new research, as provided by the present article, underlining thus the usefulness of the paper.

l. 240-242. The author states: “it is worth noting that we previously reported evidence of objective memory impairments in GWVs with subjective memory complaints [70].” This expression doesn’t seem to make sense.

l. 231-262. Historical facts and discussions about the definition and the diagnostic criteria with regard to the main clinical disorder of concern, belong to the Introduction. In the Discussion section, the author may proceed by discussing the obtained results and the reason why these results indicate the superiority of the actuarial over the conventional method. Here, the author refers to previous work only and doesn’t even mention the results of the present study (l. 255-262).

l. 289-309: These conclusions would be more accurate if regression analysis had been carried out, as mentioned earlier.

l. 340-342: As mentioned earlier, this statement doesn’t appear to be justified.

l. 345: It is not clear, to what exactly “this finding” refers.

Author Response

Reviewer's point # 1: The structure of the paper with a very short introductory section and a large discussion section, where a multitude of issues are addressed and discussed that weren’t initially mentioned in the aims of the paper, leads to some confusion about what this paper was really about.

Response: The paper has been restructured and many of the points addressed in the discussion have been moved to the introduction.

Reviewer's point # 2: The major issue of concern is the main goal of the paper (the prevalence of MCI in GWV) and the way it is displayed and discussed.

Response: The main goal of the paper has been re-framed in the last paragraph of the introduction as follows:

"Most MCI studies have focused on individuals in their 70s [36]; however, it has been argued that MCI can be identified in adults before the age of 60 using neuropsychological assessments that do not have ceiling effects and cover multiple cognitive domains [37].  We previously reported that GWVs with subjective memory complaints have objective memory impairment [38]. Because this satisfies two of the original Petersen criteria for MCI [14], and because there is suggestive evidence that GWVs exhibit a higher than expected incidence of cognitive impairment [7-10] and may be experiencing accelerated aging [6], the current study sought to investigate whether GWVs exhibit a higher than expected prevalence of MCI. We also explored demographic, clinical, military, and deployment-related differences between GWVs with and without MCI."

The discussion about the prevalence of MCI in GWVs has been restructured in the second paragraph of the discussion as follows:

"How significant is a prevalence of 12% of MCI in GWVs who had a median age of 48 years at the time they were studied? According to the American Academy of Neurology Practice update summary of MCI [80], the prevalence of MCI in the general population is 6.7% for ages 60-64, 8.4% for 65-69, 10.1% for 70-74, 14.5% for 75-59, and 25.2% for ages 80-84. Very few research studies of MCI focus on adults under the age of 60 [36]. The few studies that have reported prevalence in the range of 0%-13% [81-83]. If the higher estimates of MCI in adults under 60 is valid, then our finding that 12% of deployed GWVs had MCI is not out of the ordinary.  In a study of 1,126 twins from the Vietnam Era Twin Study of Aging (VETSA) who were 51-59 years old at the time of study, Kremen et al. reported a prevalence of 1%-65% for MCI [37]. However, it seems unlikely that 65% of Vietnam Veterans in their 50s would have MCI given that the prevalence of MCI increases with age [84] and 65% is significantly higher than the prevalence of MCI in octogenarians, estimated to be 25% by the American Academy of Neurology [80]. Consequently, Kremen et al. proposed that lower rates of MCI are more likely to be valid in adults under 60 [37]. If we assume that Kremen et al.’s proposal is correct, and that the prevalence of MCI in adults under 60 is lower than the prevalence of MCI in people ages 60-64, estimated to be 6.7% by the American Academy of Neurology [80], then 12% MCI in deployed GWVs is almost twice the prevalence of MCI expected in the general population."

Reviewer's point # 3: The author tries to access causality of MCI among GWV with regard to the effect of mental disorders and deployment-related exposure to chemicals and air pollutants. The author concludes that all these conditions are probable risk factors for MCI in GWV. This argumentation would have been more solid if the author had performed regression analysis, taking all significant covariates into account, such as demographic data. The author clearly performed regression analysis with regard to MCI and neuroimaging data, so it is a surprise that the author didn’t chose to proceed the same way here.

Response: I would like to thank the reviewer for the astute suggestion of using regression analyses to access causality of MCI among GWVs. I have added four post-hoc regression analyses to the paper:

The first examines the ability mental health disorders (i.e., current PTSD, current MDD, history of alcohol abuse/dependence; independent variables entered in the last step of the post-hoc regression model) to predict MCI status (dependent variable) over and above demographic and military characteristics that differed significantly between the groups (independent variables entered into the first step of the regression model).

The second and third post-hoc regressions examined the ability of MCI status (independent variable entered in the last step of the model) to predict hippocampal volume (dependent variable in second post-hoc analysis) and parietal cortex thickness (dependent variable in third post-hoc analysis) over and above demographic and clinical variables (i.e., age, sex, years of education, current PTSD and MDD, history of alcohol abuse/dependence, ICV, APOE ε4 and ε2 status, CDC CMI, and Khamisiyah plume exposure status, entered into the first step of the regression models).

The last post-hoc regression analysis examined the ability of all the demographic, clinical, and deployment-related exposures that differed significantly between the groups to predict MCI group status (dependent variable).

The results of the post-hoc regression are as follows:

1) The only demographic and military characteristics that differed significantly between the groups were race and rank. Post-hoc regression revealed that these variable were significantly associated with MCI status: enlisted during the GW: standardized coefficient β=0.21, t=2.99, p=0.003 and being non-white: standardized coefficient β=0.21, t=3.04, p=0.003.

Even after accounting for rank and race in the first step of a post-hoc regression, current PTSD status (standardized coefficient β=0.17, t=2.47, p=0.01) was still significantly associated with MCI status in the second step of the model. In contrast, current MDD and history of alcohol abuse/dependence were not significantly associated with MCI status.

2) After accounting for ICV (standardized coefficient β = 0.51, t= 7.40, p<0.001), age (standardized coefficient β = -0.20, t= -3.01, p=0.001), and Khamisiyah exposure status (standardized coefficient β = -0.13, t= -1.99, p<0.05), MCI status was still significantly associated with hippocampal volume (standardized coefficient β = 0.20, t= 3.07, p=0.002). Similarly, after accounting for age in the first step of the post-hoc regression (standardize coefficient β = -0.20, t= -2.44, p=0.016), MCI status was still significantly associated with parietal cortex thickness (standardized coefficient β = 0.21, t= 2.67, p=0.008).

3) In the final post-hoc analysis, I examined the relationship between demographic and military characteristics (race and rank during GW), clinical characteristics (current PTSD and MDD, history of alcohol abuse/dependence, and CDC CMI status) and deployment-related experiences (coming into contact with dead animals and witnessing living area sprayed with pesticides) that differed significantly between the groups to predict MCI status (dependent variable). The step-wise linear regression models revealed three significant predictors of MCI: race (being non-white, standardized coefficient β = 0.20, t= 2.82, p=0.005), current PTSD status (standardized coefficient β = 0.20, t= 2.94, p=0.004), and rank (enlisted during the GW, standardized coefficient β = 0.17, t= 2.48, p=0.01). After accounting for these three variables, deployment-related experiences were no longer significantly associated with MCI status.

Reviewer's point # 4: This study has more goals than to simply assess the prevalence of MCI among GWV and this has to be addressed clearly. Further, there are only few studies on the prevalence of MCI in individuals aged <65 years, which I think deserves more emphasis in the introduction, since that would underline the potential usefulness of this paper. I also recommend the use of the term “prevalence” instead of “incidence”, since the term “incidence” refers to the rate of occurrence of new cases at a given time.

Response: The term incidence has been replaced with prevalence throughout the manuscript.  The study's goals has been re-framed as follows in the last paragraph of the introduction:

"Most MCI studies have focused on individuals in their 70s [36]; however, it has been argued that MCI can be identified in adults before the age of 60 using neuropsychological assessments that do not have ceiling effects and cover multiple cognitive domains [37]. We previously reported that GWVs with subjective memory complaints have objective memory impairment [38]. Because this satisfies two of the original Petersen criteria for MCI [14], and because there is suggestive evidence that GWVs exhibit a higher than expected incidence of cognitive impairment [7-10] and may be experiencing accelerated aging [6], the current study sought to investigate whether GWVs exhibit a higher than expected prevalence of MCI. We also explored demographic, clinical, military, and deployment-related differences between GWVs with and without MCI."

Reviewer's point # 5: Classification criteria need to be described in more detail.  The author states that “Jak et al. [27] used three neuropsychological measures…”, while “Bondi et al. [28] used two neuropsychological measures…”. More information of how exactly these measures are “used” would be helpful.

Response:  I now describe the original Jak/Bondi actuarial MCI criteria more clearly. I have also included another table (Table 2) summarizing how MCI, intermediate, and cognitively normal status were operationalized in the present study.  The classification of MCI by the actuarial neuropsychological criteria has been revised in the Methods section as follows:

"The present study sought to examine the prevalence of MCI in deployed GWVs. Although we previously reported evidence that GWVs with subjective memory complaints have objective memory impairment [38], which meets two of the original Petersen criteria for MCI [14], we could not use the conventional MCI criteria to classify GWVs because the parent study did not access activities of daily living or the clinical presence/absence of dementia.  However, all participants in the parent study underwent neuropsychological assessment. Therefore, we used the actuarial neuropsychological criteria [33, 46] to determine MCI status in the GWVs. The neuropsychological measures used to classify MCI are listed in Table 1.

In the original publication by Jak et al. describing the actuarial neuropsychological criteria [33], three neuropsychological measures in five cognitive domains were used to classify MCI. A subject was classified as MCI if s/he scored > 1 SD below the age-adjusted norm on two measures in a cognitive domain (i.e., a subject with > 2 scores > 1 SD below the age-adjusted norm in the memory domain was classified as an amnestic MCI; a subject with > 2 scores > 1 SD below the age-adjusted norm in the executive function domain was classified as an executive function MCI). In a subsequent publication, Bondi et al. [35] used the actuarial neuropsychological criteria to classify subjects from the Alzheimer’s Disease Neuroimaging Initiative (ADNI). Because ADNI did not have as extensive a neuropsychological battery as that used in the original study by Jak et al. [33], the authors adapted the actuarial criteria to consider two neuropsychological measures in three cognitive domains for classifying MCI [35]. An ADNI subject was classified as MCI if s/he scored > 1 SD below the age-adjusted norm on both measures in a cognitive domain (i.e., memory, executive function, or attention) or if s/he scored > 1 SD below the age-adjusted norm on > 1 measure in each of the three cognitive domains (i.e., a subject with one score 1 SD below the age-adjusted norm in all three domains was classified as MCI).

In this study, we used the same five cognitive domains describe by Jak et al. [33]. However, because we did not have three separate neuropsychological measures for the visuospatial and language domains, we employed a hybrid of the actuarial criteria described by Jak et al. [33] and Bondi et al. [35]: We considered two neuropsychological measures in three cognitive domains (i.e., episodic memory, executive function, and attention) and one neuropsychological measure in the other two cognitive domains (i.e., language and visuospatial function). A GWV was classified as MCI if s/he scored > 1 SD below the age-adjusted norm on both measures in the episodic memory, executive function, or attention domain. A GWV was also classified as MCI if s/he scored > 1 SD below the age-adjusted norm on > 4 cognitive domains. A GWV was classified as having “intermediate” cognitive impairment if s/he scored > 1 SD below the age-adjusted norm on < 3 cognitive domains. GWVs with no score below the age-adjusted norm in any cognitive domain were classified as “cognitively normal” (CN).  See Table 2."

Reviewer's point # 6a: 15 different exposures are displayed, however only 4 appear in statistical analysis (tables 4&5). This applies of course to the correction for multiple comparisons (l. 154-155). If only 4 different exposures were taken into account, then n=4 and not 15. And what about these other 11 types of exposure?

Response: I now include all 15 exposures/experiences and their statistical significance in Table 4. There were group differences with respect to the frequency of exposure to 6 deployment-related experiences; however, only two remained significant after adjustment for multiple comparison according to the number of exposures/experiences analyzed (n=15) and the average intercorrelation among them (r=0.21) (Sankoh et al., 1997).

Reviewer's point # 6b: Some of {the exposures} refer to acute stress reactions and exposure to life threatening events, like for example, “being within 1 mile of a SCUD missile explosion”. Why didn’t the author make any comments on that? It would be very interesting to see in which degree some traumatic events predispose an individual to MCI and if this connection vanishes after controlling for PTSD.

Response: According to Dr. Lea Steele, author of the Gulf War Military History and Health Questionnaire, the question about being within 1 mile of a SCUD missile explosion was meant to estimate potential exposure to chemical and biological warfare agents as it was believed that these agents may have been weaponized in the ballistic missiles. Although there are some questions in the Gulf War Military History and Health Questionnaire that ask about stressful situations (e.g., being directly involved in ground or air combat, witnessing American or Allied troops, Iraqis or civilians badly wounded or killed), because the questions were posed in a way that was meant to estimate the duration of the exposures, the questionnaire is not the most appropriate instrument for ascertaining the traumatic nature of the events. Instead, I used the Life Stressor Checklist in the clinical evaluations to assess acute stress reactions and exposure to life threatening events in the parent study.  For that reason, and because there were no significant group differences in the duration of exposure to stressful GW experiences (e.g., being within a mile of SCUD missile explosions, being directly involved in ground or air combat, witnessing American or Allied troops, Iraqis or civilians badly wounded or killed), I only included the 15 exposures to potentially toxic substances (e.g., chemical/biological warfare agents, pesticides) queried by the Gulf War Military History and Health Questionnaire in Table 4.

Reviewer's point # 6c: Do neurophsysiological mechanisms underlying an acute stress reaction have any effect of MCI prevalence and on hippocampal atrophy later in life? Are these effects rather mediated by the neurophysiological mechanisms involved in PTSD? From a clinical point of view, I would prefer to read more about that and less about actuarial vs conventional diagnostic criteria, which seems to cover a lot of space in the present manuscript.

Response: I have expanded the discussion about the potential role of PTSD in MCI prevalence, cognitive impairment, hippocampal atrophy and risk for dementia as follows: 

"The mechanisms underlying higher rates of MCI in GWVs are likely to be complex and interrelated with no single process explaining the relationship. However, post-hoc regression analysis revealed a strong association between current PTSD and MCI status, even after accounting for demographic characteristics. PTSD is a stress-related condition that develops in some individuals after experiencing a traumatic event [85]. It has been postulated that dysregulation of the hypothalamic-pituitary-adrenal (HPA) axis occurs in some people following exposure to severe trauma [86]. Chronic hyperactivation of the HPA axis may lead to aberrant neuroimmune responses [87], which, in turn, can result in damage to the hippocampus [88-91]. A large (n=1868) consortium study recently confirmed the negative association between PTSD and hippocampal volume [91]. Notably, hippocampal atrophy is also a pathological hallmark of AD [92]. There is also suggestive evidence that PTSD may induce epigenetic changes that disrupt a number of physiological mechanisms/systems such as the metabolic, immune, and inflammatory systems, which renders individuals with PTSD vulnerable for developing a variety of co-morbid chronic diseases [93]. Aberrant immune responses may also interrupt anti-inflammatory Aβ clearance mechanisms that creates a switch towards pro-inflammatory mechanisms which could cause neuronal necrosis [94]. One consequence of this would be cognitive impairment. Indeed, PTSD has been associated with impaired cognition [95-99], increased risk for dementia [94, 100, 101], and increased Aβ burden [102], but see [103]."

Reviewer's point # 7: Table 2 needs some attention: There can’t be 7 (8%) individuals of Kansas GWI cases in the MCI group. 7 (28%) seems accurate.

Response: This was a typo. The reviewer is correct that 28% of the GWVs in the MCI group met the Kansas GWI case definition. This mistake has been corrected in the revision.

Reviewer's point # 8: With regard to ANCOVA (Table 3), 95% confidence intervals have to be provided, in order to draw additional conclusions about the reliability of the displayed results.

Response: 95% confidence intervals have been added to Table 3.

Reviewer's point # 9: l. 205-207. As mentioned above, this statement needs to be revised, since there are only 4 factors involved which have to be taken into account when adjusting to multiple comparisons.

Response: The results section describing differences in GW deployment related exposures/experiences has been re-written.  I now show all 15 exposures/experiences in Table 4 and describe in greater detail how adjustments for multiple comparisons were performed.  Finally, the revised manuscript includes a post-hoc regression analysis that examines the relationship between GW deployment-related exposures/experiences and MCI status after accounting for demographic, military, and clinical characteristics.  Deployment related-exposures/experiences were not significantly related to MCI stats after accounting for demographic, military, and clinical characteristics.

Reviewer's point # 10: l. 218-227. Paragraphs like these appear usually the Introduction, since they summarize previous work that has been accomplished, leading to argumentations about the necessity of conducting new research, as provided by the present article, underlining thus the usefulness of the paper.

Response: The manuscript has been re-organized such that summaries of previous work have been moved to the introduction.

Reviewer's point # 11: l. 240-242. The author states: “it is worth noting that we previously reported evidence of objective memory impairments in GWVs with subjective memory complaints [70].” This expression doesn’t seem to make sense.

Response: This sentence has been moved to the introduction. Hopefully, in that context, the expression makes more sense and explains the motivation for performing the study.

Reviewer's point # 12: l. 231-262. Historical facts and discussions about the definition and the diagnostic criteria with regard to the main clinical disorder of concern, belong to the Introduction. In the Discussion section, the author may proceed by discussing the obtained results and the reason why these results indicate the superiority of the actuarial over the conventional method. Here, the author refers to previous work only and doesn’t even mention the results of the present study (l. 255-262).

Response: The manuscript has been re-organized such that historical facts and discussions about the definition and the diagnostic criteria have been moved to the introduction. The discussion section has been re-framed to contain mostly discussion of the study's results.

Reviewer's point # 13: l. 289-309: These conclusions would be more accurate if regression analysis had been carried out, as mentioned earlier.

Response: The suggested regression analyses have been included as post-hoc analyses. 

Reviewer's point # 14: l. 340-342: As mentioned earlier, this statement doesn’t appear to be justified.

Response: I have re-framed the discussion about the significance of 12% of MCI among GWVs who were in their 50s at the time of testing is greater than that expected in the generation population.  See response to reviewer's point #2.

Reviewer's point # 15: l. 345: It is not clear, to what exactly “this finding” refers.

Response: The conclusion has been re-worded as follows:

"Assuming that the prevalence of MCI in people under 60 is lower than the prevalence of MCI in people ages 60-64, estimated to be 6.7% [80], the finding that 12% of GWVs (median age 48 years at the time of testing) had MCI is nearly twice the prevalence rate of MCI expected in the general population. This finding is consistent with idea that GWVs are aging at a faster rate than the general population [6]. Furthermore, GWVs with MCI had hippocampal atrophy and thinner parietal cortex, two hallmarks of AD pathogenesis [78, 154], compared to GWVs without MCI.  Because individuals with MCI develop dementia at a higher rate than the general population (10-15% versus 1-2% per year) [12], if these results are confirmed in a larger, more general sample of GWVs, it may portend higher rates of future dementia in deployed GWVs.  With the advent of in vivo biomarkers of amyloid and tau, AD is increasingly being conceptualized as a biomarker-driven diagnosis rather than a clinical syndrome [153].  Therefore, it will be informative to examine in vivo levels of amyloid and tau in GWVs with MCI in future studies."

Reviewer 2 Report

This is a well written and carefully conducted neuropsychological study of the incidence of Mild Cognitive Impairment (MCI) among Gulf War Veterans (GWV) compared to that of the general population. A convenience sample of 202 GWVs with a mean age of 54 years (age range  44-62 years, SD = 8 years) was surveyed using multiple neuropsychological measuring instruments. Controlling for biographical variables, differential toxic environment exposure during active service and prediagnosed diseases, it was found that 12% of the GWV sample (mean age range of 45-59), experienced MCI which (the authors state) was "significantly higher" than that of the general population ( age range 45-65). These are large age ranges of approximately 20 years  for MCI in both GWVs and the general population, which suggests that a statistical comparison of median values may give additional strength to the research analysis and counteract the inherent danger in a post-test only type research design of making a Type 1 error that the study is exposed to. The fact that the study is non-random non-experimental and without pre-test and post-test (with control group) scores of neuropsychological functioning among GW service personnel, is a limitation to the validity of the causal inferences and general overall  conclusions made in the study. i believe the the authors need to discuss this aspect in more detail in the conclusion and limitations section of the paper. Also, the fact that 12% of GWV were diagnosed as MCI needs to be considered against the indicated range of MCI in the general population, stated in the paper as 0-13%. Does this unequivocally support the authors' conclusion that 12% MCI found in GWV is significantly higher than that of the general population? While the authors should be congratulated on producing the comprehensive and detailed analysis presented in the paper, its research design weaknesses and the danger of making a Type 1 error need to be further discussed and elborated in the conclusion.

There is  minor additional point that the authors might consider clarifying.  Under Table 3, line 200 of the manuscript, the authors' state: "p=0.04, not significant after multiple comparisons". This is not clear,  as it stands it indicates significance at the <0.05 level. what was p. after the multiple comparisons (and with what)?     

Author Response

Reviewer's point #1: These are large age ranges of approximately 20 years  for MCI in both GWVs and the general population, which suggests that a statistical comparison of median values may give additional strength to the research analysis and counteract the inherent danger in a post-test only type research design of making a Type 1 error that the study is exposed to. 

Response: I now report the median age of the entire cohort (52 years) and that of the GWVs with MCI (48 years) in the abstract. In response to Reviewer 1, I have re-framed the discussion about the significance of the prevalence of 12% MCI in deployed GW Veterans in the discussion.

Reviewer's point #2: The fact that the study is non-random non-experimental and without pre-test and post-test (with control group) scores of neuropsychological functioning among GW service personnel, is a limitation to the validity of the causal inferences and general overall  conclusions made in the study. I believe the the authors need to discuss this aspect in more detail in the conclusion and limitations section of the paper. 

Response: I now cite the fact that the study has a cross-sectional, non-random non-experimental and without pre-test and post-test (with control group) scores of neuropsychological functioning among GW service personnel as a limitation in the discussion section of the paper.  I also point out that the findings of this study may not accurately reflect the true prevalence or severity of MCI among GWVs and require replication in a larger, more general sample of GWVs.

Reviewer's point #3: The fact that 12% of GWV were diagnosed as MCI needs to be considered against the indicated range of MCI in the general population, stated in the paper as 0-13%. Does this unequivocally support the authors' conclusion that 12% MCI found in GWV is significantly higher than that of the general population? 

Response: Reviewer 1 brought up a similar critique and I have tried to address this point in the discussion section as follows:

"How significant is a prevalence of 12% of MCI in GWVs who had a median age of 48 years at the time they were studied? According to the American Academy of Neurology Practice update summary of MCI [80], the prevalence of MCI in the general population is 6.7% for ages 60-64, 8.4% for 65-69, 10.1% for 70-74, 14.5% for 75-59, and 25.2% for ages 80-84. Very few research studies of MCI focus on adults under the age of 60 [36]. The few studies that have reported prevalence in the range of 0%-13% [81-83]. If the higher estimates of MCI in adults under 60 is valid, then our finding that 12% of deployed GWVs had MCI is not out of the ordinary.  In a study of 1,126 twins from the Vietnam Era Twin Study of Aging (VETSA) who were 51-59 years old at the time of study, Kremen et al. reported a prevalence of 1%-65% for MCI [37]. However, it seems unlikely that 65% of Vietnam Veterans in their 50s would have MCI given that the prevalence of MCI increases with age [84] and 65% is significantly higher than the prevalence of MCI in octogenarians, estimated to be 25% by the American Academy of Neurology [80]. Consequently, Kremen et al. proposed that lower rates of MCI are more likely to be valid in adults under 60 [37]. If we assume that Kremen et al.’s proposal is correct, and that the prevalence of MCI in adults under 60 is lower than the prevalence of MCI in people ages 60-64, estimated to be 6.7% by the American Academy of Neurology [80], then 12% MCI in deployed GWVs is almost twice the prevalence of MCI expected in the general population."

Reviewer 3 Report

This manuscript was well written. I do think reading through the paper again for spelling and grammatical errors would be wise. Some section are also not as smooth in reading as they could be, but overall I think the study is sound methodologically and reasonable in the authors claims. 

I did find myself interested if the authors compared different branches of the military and found any significant differences between groups. I do not think this takes away from the study, but something I was wondered. 

Author Response

Reviewer's point #1: I do think reading through the paper again for spelling and grammatical errors would be wise. Some section are also not as smooth in reading as they could be.

Response: I have proof-read the manuscript for spelling, grammatical errors and revised it to be more clear and easy to comprehend.

Reviewer's point #2: I did find myself interested if the authors compared different branches of the military and found any significant differences between groups.

Response: There were no group differences in memberships in the different branches of the military. This information is provided in Table 2.

Reviewer 4 Report

Overall, I do not have any huge issues with this paper. It is interesting, though difficult to understand. 

First, I think that the method section needs to be made clearer - especially the 'correction for multiple comparisons'. For me personally, it is not clear what this section shows, or why 'a 2-sided adjusted p=0.006' was considered significant, when in the table the normal 0.05, 0.01 and 0.001 rates are still used. Though, I'm sure this is mainly due my inexperience with such methods.

Second, in the discussion, and in general, I sometimes have the feeling you are making the link between MCI and Khamisyah plume too easily. Your paper is cross-sectional - this is also a limitation that really should be made more clearly in your limitation-part - and as such, making such causal relations is odd. Especially considering that your sample is now in their 70s and these people are Gulf-war veterans, so: there are decades between this measurement and the war. Generally speaking, people with PTSD and such, will have a higher chance of doing things that inflict more PTSD-symptoms. It is possible that the MCI is also related to many other factors, perhaps much more relevant and true. We don't know. It's just an association. An important one, yes, but still, an association, not a causal link. 

Third, I would make much clearer what this study shows that wasn't yet known. Your discussion begins basically with a whole heap of other data, but what does this study really show then? It is an important study, but I really would like to see a paragraph where you just state: this is what this study shows which wasn't yet known. 

Fourth, I think the whole situation with that there is also another study using the same data needs to be much clearer. Because at the end I wasn't sure how the two papers were seperated. This is especially noteble in the measures-section, where senteces such '...in the original study...' are quite confusing. I would recommend that you write what the present study does, and then you can add at the end 'which is the same as in study X'. Because now I am confused as to what your study will use. Because, yeah, the original study used X and Y, but what do you do in THIS study. 

Fifth, aligned with that, you really should say something more about the ethical aspect of this. Because, if I understand correctly, you are using data that was not meant to be used, right? Because, if you follow it stricktly, do you have any approval for this study? Please say something more about it, because again, it only says that your original study was approved. The only thing I actually want to know about, is everything about the study I'm reading - because I'm not going to look up the previous study to understand the present one. 

Finally, some small idiomatic uses of language which might be confusing for non-English speakers. The one that I think should be corrected is 'In this spirit', line 49. I think that this is an idiom (I could be wrong) and as such, it might be not appropriate for an international paper. 

But, overall, an interesting paper, and as you can see, I do not have a lot of critique. 

Author Response

Reviewer's point #1: I think that the method section needs to be made clearer - especially the 'correction for multiple comparisons'. For me personally, it is not clear what this section shows, or why 'a 2-sided adjusted p=0.006' was considered significant, when in the table the normal 0.05, 0.01 and 0.001 rates are still used. 

Response: I have re-framed the description of the corrections for multiple comparisons to be more clear in the method and results section.  Briefly, Bonferroni correction for multiple comparisons can be overly conservative when applied to non-independent (i.e. correlated) measures. For this reason, I adjusted the analyses for multiple comparisons according to the number of correlated measures examined and the average intercorrelations among the measures, as described by Sankoh et al., (1997).  The tables include the actual p-values. However, the p-values that remained significant after adjustments for multiple comparisons are highlighted in bold text.

Reviewer's point #2: In the discussion, and in general, I sometimes have the feeling you are making the link between MCI and Khamisyah plume too easily. Your paper is cross-sectional - this is also a limitation that really should be made more clearly in your limitation-part - and as such, making such causal relations is odd. Especially considering that your sample is now in their 70s and these people are Gulf-war veterans, so: there are decades between this measurement and the war. Generally speaking, people with PTSD and such, will have a higher chance of doing things that inflict more PTSD-symptoms. It is possible that the MCI is also related to many other factors, perhaps much more relevant and true. We don't know. It's just an association. An important one, yes, but still, an association, not a causal link.

Response: I now state in the limitations section of the discussion:

"The findings of this study should be considered in the context of several limitations: First, this study had a cross-sectional, non-random, non-experimental design because the parent study from which data for the secondary analyses were derived was not originally designed to investigate the relationship between GW deployment and the prevalence of MCI. Thus, the current findings may not accurately reflect the true prevalence or severity of MCI among GWVs and cannot determine causal links."

In response to Reviewer 1's comments, I have also added several post-hoc regression analyses to examine the relationship between MCI status and other variables (e.g., demographic and clinical) that differed significantly between the groups. 

Reviewer's point #3: I would make much clearer what this study shows that wasn't yet known. Your discussion begins basically with a whole heap of other data, but what does this study really show then? It is an important study, but I really would like to see a paragraph where you just state: this is what this study shows which wasn't yet known. 

Response: In response to comments from Reviewers 1 and 2, I have re-framed the discussion to about the significance of finding that 12% of GWVs who were a median of 48 years at the time of testing met the criteria for MCI. Hopefully this makes the point more clear. I have also stated more explicitly in the introduction what the goals of the study are:

"Most MCI studies have focused on individuals in their 70s [36]; however, it has been argued that MCI can be identified in adults before the age of 60 using neuropsychological assessments that do not have ceiling effects and cover multiple cognitive domains [37]. We previously reported that GWVs with subjective memory complaints have objective memory impairment [38]. Because this satisfies two of the original Petersen criteria for MCI [14], and because there is suggestive evidence that GWVs exhibit a higher than expected incidence of cognitive impairment [7-10] and may be experiencing accelerated aging [6], the current study sought to investigate whether GWVs exhibit a higher than expected prevalence of MCI. We also explored demographic, clinical, military, and deployment-related differences between GWVs with and without MCI."

Reviewer's point #4: I think the whole situation with that there is also another study using the same data needs to be much clearer. Because at the end I wasn't sure how the two papers were seperated. This is especially noteble in the measures-section, where senteces such '...in the original study...' are quite confusing. I would recommend that you write what the present study does, and then you can add at the end 'which is the same as in study X'. Because now I am confused as to what your study will use. Because, yeah, the original study used X and Y, but what do you do in THIS study. 

Response: This point was also raised by Reviewers 1 and 2. Consequently, I have revised the methods section to more clearly describe the actuarial MCI criteria, as it was originally proposed by Jak and Bondi, and how it was operationalized in the present paper.  I have also included a new table (Table 2) summarizing how the actuarial MCI criteria was operationalized in the present paper. Finally, I have eliminated the terms "original paper" and "original study" to avoid confusion about the parent study from which data for the current secondary analyses originated and the papers by Jak et al. (2009) and Bondi et al. (2014) describing the actuarial MCI criteria and adaptation of the actuarial MCI criteria for the Alzheimer's Disease Neuroimaging Initiative (ADNI) dataset.

Reviewer's point #5: You really should say something more about the ethical aspect of this. Because, if I understand correctly, you are using data that was not meant to be used, right? Because, if you follow it strictly, do you have any approval for this study? Please say something more about it, because again, it only says that your original study was approved. The only thing I actually want to know about, is everything about the study I'm reading - because I'm not going to look up the previous study to understand the present one. 

Response: I now clarify in the Human Subjects section that this study consisted of secondary analyses of pre-existing de-identified data. For that reason, the UCSF IRB did not require additional approval for the secondary analyses.  Moreover, the consent form of the parent study included language informing participants that de-identified may be subject to secondary analyses after completion of the study.

Reviewer's point #6: Some small idiomatic uses of language which might be confusing for non-English speakers. The one that I think should be corrected is 'In this spirit', line 49. I think that this is an idiom (I could be wrong) and as such, it might be not appropriate for an international paper. 

Response: The term "in this spirit" has been removed from the manuscript.

Round 2

Reviewer 1 Report

The manuscript was improved significantly. I agree with the current version.